# Context Transformer with Stacked Pointer Networks for Conversational Question Answering over Knowledge Graphs

Joan Plepi[1][*][†], Endri Kacupaj[2][*], Kuldeep Singh[3], Harsh Thakkar[4], and Jens Lehmann[2,5]

[1] Technische Universität Darmstadt, Darmstadt, Germany
joan.plepi@tu-darmstadt.de
[2] Smart Data Analytics Group, University of Bonn, Bonn, Germany
{kacupaj,jens.lehmann}@cs.uni-bonn.de
[3] Zerotha Research and Cerence GmbH, Germany
kuldeep.singh1@cerence.com
[4] Zerotha Research and Osthus GmbH, Germany
harsh.thakkar@osthus.com
[5] Fraunhofer IAIS, Dresden, Germany
jens.lehmann@iais.fraunhofer.de

**Abstract.** Neural semantic parsing approaches have been widely used for Question Answering (QA) systems over knowledge graphs. Such methods provide the flexibility to handle QA datasets with complex queries and a large number of entities. In this work, we propose a novel framework named CARTON (**C**ontext tr**A**nsforme**R** s**T**acked p**O**inter **N**etworks), which performs multi-task semantic parsing for handling the problem of conversational question answering over a large-scale knowledge graph. Our framework consists of a stack of pointer networks as an extension of a context transformer model for parsing the input question and the dialog history. The framework generates a sequence of actions that can be executed on the knowledge graph. We evaluate CARTON on a standard dataset for complex sequential question answering on which CARTON outperforms all baselines. Specifically, we observe performance improvements in F1-score on eight out of ten question types compared to the previous state of the art. For logical reasoning questions, an improvement of 11 absolute points is reached.

**Keywords:** Conversational Question Answering · Knowledge Graph · Context Transformer · Stacked Pointer Networks.

## 1 Introduction

Recently, there has been an increased demand for chatbots and voice assistants to communicate and assist humans in different domains such as chitchat, medical, news, enterprise, etc ([7]). Question answering is a common phenomenon in chatbot conversations to seek specific information. While such questions inherit a conversational context, humans also tend to ask questions that require complex reasoning to answer in a real-world scenario. The complexity of questions may differ at various granularity (e.g., simple,

---

[*] Denotes equal contribution to this research
[†] Work was done while the author was a student at University of Bonn

**Table 1.** An example dialog from CSQA dataset [19] describing various states of a conversation.

| Turn | State | Utterance |
| --- | --- | --- |
| T1 | Simple Question | **USER**: Can you tell me which cities border Verderio Inferiore ? |
|  | Response | **SYSTEM**: Cornate d'Adda, Bernareggio, Robbiate |
| T2 | Coreference Question | **USER**: And which cities flank that one ? |
|  | Clarification | **SYSTEM**: Did you mean Robbiate? |
| T3 | Clarification Question | **USER**: No, I meant Cornate d'Adda. |
|  | Response | **SYSTEM**: Bottanuco, Busnago, Trezzo sull'Adda |
| T4 | Logical Question | **USER**: Which ones share border with Robbiate and are not part of Azerbaijan? |
|  | Response | **SYSTEM**: Trezzo sull'Adda, Busnago, Bottanuco |

logical, quantitative, and comparative). Table 1 presents a few examples from a complex question answering dataset with a conversational context [19]. The example dialogue has several question types and challenges. For example, in the first turn, the user asks a simple direct question, and in the following turn, she asks a question that refers to the context from the previous turn. Furthermore, in the last turn, there is a question that requires logical reasoning to offer a multitude of complexity. Given these questions are from the general knowledge domain, the information required to answer questions can be extracted from publicly available large-scale Knowledge Graphs (KGs) such as DBpedia [11], Freebase [1], and Wikidata [27].

Neural semantic parsing approaches for question answering over KGs have been widely studied in the research community [9,13,8,5]. In a given question, these approaches use a semantic parsing model to produce a logical form which is then executed on the KG to retrieve an answer. While traditional methods tend to work on small KGs [28], more recent approaches also work well on large-scale KGs [8,21]. Often, researchers targeting large scale KGs focus on a stepwise method by first performing entity linking and then train a model to learn the corresponding logical form for each question type [4,8]. Work in [21] argues that the stepwise approaches have two significant issues. First, errors in upstream subtasks (e.g., entity detection and linking, predicate classification) are propagated to downstream ones (e.g., semantic parsing), resulting in accumulated errors. For example, case studies in previous works [29,8,4] show that entity linking error is one of the significant errors leading to the wrong results in the question-answering task. Second, when models for the subtasks are learned independently, the supervision signals cannot be shared among the models for mutual benefits. To mitigate the limitations of the stepwise approach, [21] proposed a multi-task learning framework where a pointer-equipped semantic parsing model is designed to resolve coreference in conversations, and intuitively, empower joint learning with a type-aware entity detection model. The framework combines two objectives: one for semantic parsing and another for entity detection. However, the entity detection model uses supervision signals only

from the contextual encoder, and no further signal is provided from the decoder or the semantic parsing task.

In this paper, we target the problem of conversational (complex) question answering over large-scale knowledge graph. We propose CARTON (**C**ontext tr**A**nsforme**R** s**T**acked p**O**inter **N**etworks)- a multi-task learning framework consisting of a context transformer model extended with a stack of pointer networks for multi-task neural semantic parsing. Our framework handles semantic parsing using the context transformer model while the remaining tasks such as type prediction, predicate prediction, and entity detection are handled by the stacked pointer networks. Unlike [21] which is current state-of-the-art, CARTON's stacked pointer networks incorporate knowledge graph information for performing any reasoning and does not rely only on the conversational context. Moreover, pointer networks provide the flexibility for handling out-of-vocabulary [26] entities, predicates, and types that are unseen during training. Our ablation study 5.1 further supports our choices. In contrast with the current state of the art, another significant novelty is that the supervision signals in CARTON propagate in sequential order, and all the components use the signal forwarded from the previous components. To this end, we make the following contributions in the paper:

– CARTON - a multi-task learning framework for conversational question answering over large scale knowledge graph.
– For neural semantic parsing, we propose a reusable grammar that defines different logical forms that can be executed on the KG to fetch answers to the questions.

CARTON achieves new state of the art results on eight out of ten question types from a large-scale conversational question answering dataset. We evaluate CARTON on the Complex Sequential Question Answering (CSQA) [19] dataset consisting of conversations over linked QA pairs. The dataset contains 200K dialogues with 1.6M turns, and over 12.8M entities. Our implementation, the annotated dataset with proposed grammar, and results are on a public github[6]. The rest of this article is organized as follows: Section 2 summarizes the related work. Section 3 describes the CARTON framework. Section 4 explains the experimental settings and the results are reported in Section 5. Section 6 concludes this article.

## 2   Related Work

**Semantic Parsing and Multi-task Learning Approaches**  Our work lies in the areas of semantic parsing and neural approaches for question answering over KGs. Works in [6,15,31,16] use neural approaches to solve the task of QA. [15] introduces an approach that splits the question into spans of tokens to match the tokens to their respective entities and predicates in the KG. The authors merge the word and character-level representation to discover better matching in entities and predicates. Candidate subjects are generated based on n-grams matching with words in the question, and then pruned based on predicted predicates. However, their experiments are focused on simple questions. [31] propose a probabilistic framework for QA systems and experiment on a

---

[6] https://github.com/endrikacupaj/CARTON

new benchmark dataset. The framework consists of two modules. The first one model the probability of the topic entity $y$, constrained on the question. The second module reasons over the KG to find the answer $a$, given the topic entity $y$ which is found in the first step and question $q$. Graph embeddings are used to model the subgraph related to the question and for calculating the distribution of the answer depended from the question $q$ and the topic $y$. [13] introduce neural symbolic machine (NSM), which contains a neural sequence-to-sequence network referred also as the "programmer", and a symbolic non-differentiable LISP interpreter ("computer"). The model is extended with a key-value memory network, where keys and values are the output of the sequence model in different encoding or decoding steps. The NSM model is trained using the REINFORCE algorithm with weak supervision and evaluated on the WebQuestionsSP dataset [30]. [8] also present an approach that maps utterances to logical forms. Their model consists of a sequence-to-sequence network, where the encoder produces the embedding of the utterance, and the decoder generates the sequence of actions. Authors introduce a dialogue memory management to handle the entities, predicates, and logical forms are referred from a previous interaction. Finally, MaSP [21] present a multi-task model that jointly learns type-aware entity detection and pointer equipped logical form generation using a semantic parsing approach. Our proposed framework is inspired by them; however, we differ considerably on the following points: 1) CARTON's stacked pointer networks incorporate knowledge graph information for performing reasoning and do not rely only on the conversational context as MaSP does. 2) The stacked pointer network architecture is used intentionally to provide the flexibility for handling out-of-vocabulary entities, predicates, and types that are unseen during training. The MaSP model does not cover out-of-vocabulary knowledge since the model was not intended to have this flexibility. 3) CARTON's supervision signals are propagated in sequential order, and all the components use the signal forwarded from the previous component. 4) We employ semantic grammar with new actions for generating logical forms. While [21] employs almost the same grammar as [8].

**Other Approaches** There has been extensive research for task-oriented dialog systems such as [10] that induces joint text and knowledge graph embeddings to improve task-oriented dialogues in the domains such as restaurant and flight booking. Work present in [2] proposes another dataset, "ConvQuestions" for conversations over KGs along with an unsupervised model. Some other datasets for conversational QA include CANARD and TREC CAsT [24]. Overall, there are several approaches proposed for conversational QA, and in this paper, we closely stick to multi-task learning approaches for CARTON's comparison and contributions.

## 3   CARTON

Our focus is on conversations containing complex questions that can be answered by reasoning over a large-scale KG. The training data consists of utterances $u$ and the answer label $a$. We propose a semantic parsing approach, where the goal is to map the utterance $u$ into a logical form $z$, depending on the conversation context. A stack of three pointer networks is used to fill information extracted from the KG. The final

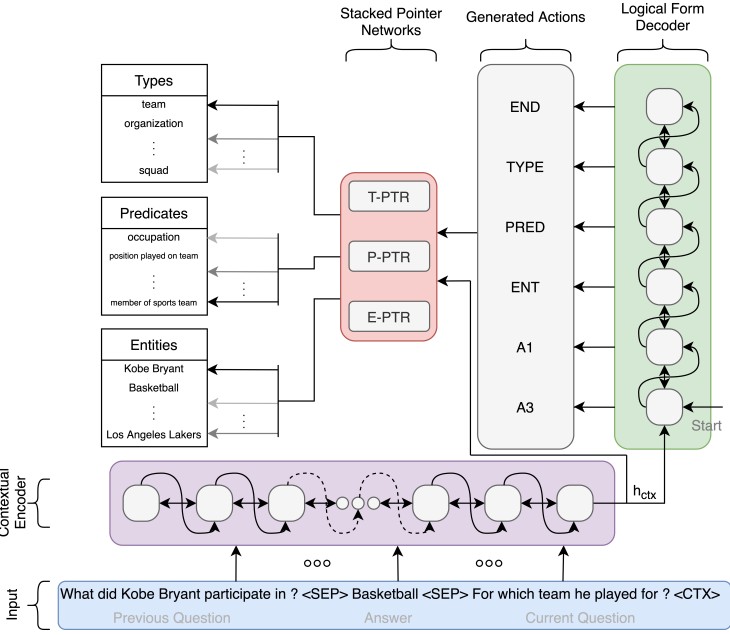

**Fig. 1.** Context Transformer with Stacked Pointer Networks architecture (CARTON). It consists of three modules: 1) A Transformer-based contextual encoder finds the representation of the current context of the dialogue. 2) A logical decoder generates the pattern of the logical forms defined in Table 2. 3) The stacked pointer network initializes the KG items to fetch the correct answer.

generated logical form aims to fetch the correct answer once executed on the KG. Figure 1 illustrates the overall architecture of CARTON framework.

### 3.1   Grammar

We predefined a grammar with various actions as shown in Table 2 which can result in different logical forms that can be executed on the KG. Our grammar definition is inspired by [8] which MaSP [21] also employs. However, we differ in many semantics of the actions and we even defined completely new actions. For example, *find* action is split into *find(e, p)* that corresponds to finding an edge with predicate *p* to the subject *e*; and *find_reverse(e, p)* finds an edge with predicate *p* with object *e*. Moreover, *per_type* is not defined by [8] in their grammar. Table 3 indicates some (complex) examples from CSQA dataset [19] with gold logical form annotations using our predefined grammar. Following [14], each action definition is represented by a function that is executed on the KG, a list of input parameters, and a semantic category that corresponds to the output of the function. For example, *set → find(e, p)*, it has a *set* as a semantic category, a function *find* with input parameters *e*, *p*. We believe that the defined actions are sufficient for creating sequences that cover complex questions and we provide empirical evidences in Section 5. Every action sequence can be parsed into a tree, where the model recursively writes

**Table 2.** Predefined grammar with respective actions to generate logical forms.

| Action | Description |
|---|---|
| set $\rightarrow$ find($e$, $p$) | set of objects (entities) with subject $e$ and predicate $p$ |
| set $\rightarrow$ find_reverse($e$, $p$) | set of subjects (entities) with object $e$ and predicate $p$ |
| set $\rightarrow$ filter_by_type(set, tp) | filter the given set of entities based on the given type |
| set $\rightarrow$ filter_mult_types($set_1$, $set_2$) | filter the given set of entities based on the given set of types |
| boolean $\rightarrow$ is_in(set, entity) | check if the entity is part of the set |
| boolean $\rightarrow$ is_subset ($set_1$, $set_2$) | check if $set_2$ is subset of $set_1$ |
| number $\rightarrow$ count(set) | count the number of elements in the set |
| dict $\rightarrow$ per_type(p, $tp_1$, $tp_2$) | extracts a dictionary, where keys are entities of $type_1$ and values are the number of objects of $type_2$ related with $p$ |
| dict $\rightarrow$ per_type_rev(p, $tp_1$, $tp_2$) | extracts a dictionary, where keys are entities of $type_1$ and values are the number of subjects of $type_2$ related with $p$ |
| set $\rightarrow$ greater(num, dict) | set of entities that have greater count than *num* |
| set $\rightarrow$ lesser(num, dict) | set of entities that have lesser count than *num* |
| set $\rightarrow$ equal(num, dict) | set of entities that have equal count with *num* |
| set $\rightarrow$ approx(num, dict) | set of entities that have approximately same count with *num* |
| set $\rightarrow$ argmin(dict) | set of entities that have the most count |
| set $\rightarrow$ argmax(dict) | set of entities that have the least count |
| set $\rightarrow$ union($set_1$, $set_2$) | union of $set_1$ and $set_2$ |
| set $\rightarrow$ intersection($set_1$, $set_2$) | intersection of $set_1$ and $set_2$ |
| set $\rightarrow$ difference($set_1$, $set_2$) | difference of $set_1$ and $set_2$ |

the leftmost non-terminal node until the whole tree is complete. The same approach is followed to execute the action sequence, except that the starting point is the tree leaves.

### 3.2   Context Transformer

The section describes the semantic parsing part of CARTON, which is a context transformer. The transformer receives input a conversation turn that contains the context of the interaction and generates a sequence of actions. Formally, an interaction $I$ consists of the question $q$ that is a sequence $x = \{x_1, \ldots, x_n\}$, and a label $l$ that is a sequence $y = \{y_1, \ldots, y_m\}$. The network aims to model the conditional probability $p(y|x)$.

**Contextual Encoder**  In order to cope with coreference and ellipsis phenomena, we require to include the context from the previous interaction in the conversation turn. To accomplish that, the input to the contextual encoder is the concatenation of three utterances from the dialog turn: 1) the previous question, 2) the previous answer, and 3) the current question. Every utterance is separated from one another using a $< SEP >$ token. A special context token $< CTX >$ is appended at the end where the embedding of this utterance is used as the semantic representation for the entire input question. Given an utterance $q$ containing $n$ words $\{w_1, \ldots, w_n\}$, we use GloVe [18] to embed the words into a vector representation space of dimension $d_{emb}$. More specifically, we

**Table 3.** Examples from the CSQA dataset [19], annotated with gold logical forms.

| Question Type | Question | Logical Forms |
| --- | --- | --- |
| Simple Question (Direct) | Q1: Which administrative territory is the birthplace of Antonio Reguero ? | filter_type( find(Antonio Reguero, place of birth), administrative territorial entity) |
| Simple Question (Ellipsis) | Q1: Which administrative territories are twin towns of Madrid ? A1: Prague, Moscow, Budapest Q2: And what about Urban Community of Brest? | filter_type( find(Urban Community of Brest, twinned administrative body), administrative territorial entity) |
| Simple Question (Coreferenced) | Q1: What was the sport that Marie Pyko was a part of ? A1: Association football Q2: Which political territory does that person belong to ? | filter_type( find(Marie Pyko, country of citizenship), political territorial entity) |
| Quantitative Reasoning (Count) (All) | Q1: How many beauty contests and business enterprises are located at that city ? A1: Did you mean Caracas? Q2: Yes | count(union( filter_type( find_reverse( Caracas, located in), beauty contest), filter_type( find_reverse(Caracas, located in), business enterprises))) |
| Quantitative Reasoning (All) | Q1; Which political territories are known to have diplomatic connections with max number of political territories ? | argmax( per_type( diplomatic relation, political territorial entity, political territorial entity)) |
| Comparative Reasoning (Count) (All) | Q1: How many alphabets are used as the scripts for more number of languages than Jawi alphabet ? | count(greater(count( filter_type(find(Jawi alphabet, writing system), language)), per_type(writing system, alphabet, language))) |
| Comparative Reasoning (All) | Q1: Which occupations were more number of publications and works mainly about than composer ? | greater(filter_type( find(composer, main subject), occupations), and( per_type(main subject, publications, occupations), per_type(main subject, work, occupations))) |
| Verification | Q1: Was Geir Rasmussen born at that administrative territory ? | is_in( find(Geir Rasmussen, place of birth), Chicago) |

get a sequence $x = \{x_1, \ldots, x_n\}$ where $x_i$ is given by,

$$x_i = GloVe(w_i)$$

and $x_i \in \mathbb{R}^{d_{emb}}$. Next, the word embeddings $x$, are forwarded as input to the contextual encoder, that uses the multi-head attention mechanism from the Transformer network [25]. The encoder outputs the contextual embeddings $h = \{h_1, \ldots, h_n\}$, where $h_i \in \mathbb{R}^{d_{emb}}$, and it can be written as:

$$h = encoder(x; \theta^{(enc)})$$

where $\theta^{(enc)}$ are the trainable parameters of the contextual encoder.

**Logical Form Decoder**  For the decoder, we likewise utilize the Transformer architecture with a multi-head attention mechanism. The decoder output is dependent on contextual embeddings $h$ originated from the encoder. The decoder detects each action and general semantic object from the KG, i.e., the decoder predicts the correct logical form, without specifying the entity, predicate, or type. Here, the decoder vocabulary consists of $V = \{A_0, A_1, \ldots, A_{18}, entity, predicate, type\}$ where $A_0, A_1, \ldots, A_{18}$ are the short names of actions in Table 2. The goal is to produce a correct logical form sequence. The decoder stack is a transformer model supported by a linear and a softmax layer to estimate the probability scores, i.e., we can define it as:

$$s^{(dec)} = decoder(h; \theta^{(dec)}), \; p_t = softmax(\boldsymbol{W}^{(dec)} s_t^{(dec)}) \tag{1}$$

where $s_t^{(dec)}$ is the hidden state of the decoder in time step $t$, $\theta^{(dec)}$ are the model parameters, $\boldsymbol{W}^{(dec)} \in \mathbb{R}^{|V| \times d_{emb}}$ are the weights of the feed-forward linear layer, and $p_t \in \mathbb{R}^{|V|}$ is the probability distribution over the decoder vocabulary for the output token in time step $t$.

### 3.3   Stacked Pointer Networks

As we mentioned, the decoder only outputs the actions without specifying any KG items. To complete the logical form with instantiated semantic categories, we extend our model with an architecture of stacked pointer networks [26]. The architecture consists of three-pointer networks and each one of them is responsible for covering one of the major semantic categories (types, predicates, and entities) required for completing the final executable logical form against the KG.

   The first two pointer networks of the stack are used for predicates and types semantic category and follow a similar approach. The vocabulary and the inputs are the entire predicates and types of the KG. We define the vocabularies, $V^{(pd)} = \{r_1, \ldots, r_{n_{pd}}\}$ and $V^{(tp)} = \{\tau_1, \ldots, \tau_{n_{tp}}\}$, where $n_{pd}$ and $n_{tp}$ is the total number of predicates and types in the KG, respectively. To compute the pointer scores for each predicate or type candidate, we use the current hidden state of the decoder and the context representation. We model the pointer networks with a feed-forward linear network and a softmax layer. We can define the type and predicate pointers as:

$$p_t^{(pd)} = softmax(\boldsymbol{W}_1^{(pd)} v_t^{(pd)}), \; p_t^{(tp)} = softmax(\boldsymbol{W}_1^{(tp)} v_t^{(tp)}), \tag{2}$$

where $p_t^{(pd)} \in \mathbb{R}^{|V^{(pd)}|}$ and $p_t^{(tp)} \in \mathbb{R}^{|V^{(tp)}|}$ are the probability distributions over the predicate and type vocabularies respectively. The weight matrices $\boldsymbol{W}_1^{(pd)}$, $\boldsymbol{W}_1^{(tp)} \in \mathbb{R}^{1 \times d_{kg}}$. Also, $v_t$ is a joint representation that includes the knowledge graph embeddings, the context and the current decoder state, computed as:

$$
\begin{aligned}
v_t^{(pd)} &= tanh(\boldsymbol{W}_2^{(pd)}[s_t; h_{ctx}] + r), \\
v_t^{(tp)} &= tanh(\boldsymbol{W}_2^{(tp)}[s_t; h_{ctx}] + \tau),
\end{aligned}
\tag{3}
$$

where the weight matrices $\boldsymbol{W}_2^{(pd)}$, $\boldsymbol{W}_2^{(tp)} \in \mathbb{R}^{d_{kg} \times 2d_{emb}}$, transform the concatenation of the current decoder state $s_t$ with the context representation $h_{ctx}$. We denote with $d_{kg}$ the dimension used for knowledge graph embeddings. $r \in \mathbb{R}^{d_{kg} \times |V^{(pd)}|}$ are the predicate embeddings and $\tau \in \mathbb{R}^{d_{kg} \times |V^{(tp)}|}$ are the type embeddings. $tanh$ is the non-linear layer. Please note, that the vocabulary of predicates and types is updated during evaluation, hence the choice of pointer networks.

The third pointer network of the stack is responsible for the entity prediction task. Here we follow a slightly different approach due to the massive number of entities that the KG may have. Predicting a probability distribution over KG with a considerable number of entities is not computationally feasible. For that reason, we decrease the size of entity vocabulary during each logical form prediction. In each conversation, we predict a probability distribution **only** for the entities that are part of the context. Our entity "memory" for each conversation turn involves entities from the previous question, previous answer, and current question. The probability distribution over the entities is then calculated in the same way as for predicates and types where the softmax is:

$$
p_t^{(ent)} = softmax(\boldsymbol{W}_1^{(ent)} v_t^{(ent)}),
\tag{4}
$$

where $p_t^{(ent)} \in \mathbb{R}^{|V_k^{(ent)}|}$, and $V_k^{(ent)}$ is the set of entities for the $k^{th}$ conversation turn. The weight matrix $\boldsymbol{W}_1^{(ent)} \in \mathbb{R}^{1 \times d_{kg}}$ and the vector $v_t$ is then computed following the same equations as before:

$$
v_t^{(ent)} = tanh(\boldsymbol{W}_2^{(ent)}([s_t; h_{ctx}]) + e_k)
\tag{5}
$$

where $e_k$ is the sequence of entities for the $k^{th}$ conversation turn. In general, the pointer networks are robust to handle a different vocabulary size for each time step [26]. Moreover, given the knowledge graph embeddings, our stacked pointer networks select the relevant items from the knowledge graph depending on the conversational context. In this way, we incorporate knowledge graph embeddings in order to perform any reasoning and do not rely only on utterance features. Furthermore, the [21] utilizes a single pointer network that only operates on the input utterance to select the already identified entities. Our stacked pointer networks do not use the input utterance but rather directly rely on the knowledge graph semantic categories (types, predicates, and entities).

### 3.4  Learning

For each time step, we have four different predicted probability distributions. The first is the decoder output over the logical form's vocabulary, and the three others from the

stacked pointer networks for each of the semantic categories (entity, predicate, and type). Finally, we define CARTON loss function as:

$$Loss_t = -\frac{1}{m} \sum_{i=1}^{m} \left( log\ p_{i\ [i=y_i^{(t)}]} + \sum_{c \in \{ent,pd,tp\}} I_{[y_i^{(t)}=c]}\ logp^{(c)}_{i\ [i=y_i^{(c_t)}]} \right), \quad (6)$$

where $Loss_t$ is the loss function computed for the sequence in time step $t$, $m$ is the length of the logical form, $y^{(t)}$ is the gold sequence of logical form, and $y^{(c_t)}$ is the gold label for one of the semantic categories $c \in \{ent, pd, tp\}$.

## 4   Experimental Setup

**Dataset and Experiment Settings**  We conduct our experiments on the Complex Sequential Question Answering (CSQA) dataset[7] [19]. CSQA was built on the Wikidata KG. The CSQA dataset consists of around 200K dialogues where each partition train, valid, test contains 153K, 16K, 28K dialogues, respectively. The questions in the CSQA dataset involve complex reasoning on Wikidata to determine the correct answer. The different question types that appear in the dataset are simple questions, logical reasoning, verification, quantitative reasoning, comparative reasoning, and clarification. We can have different subtypes for each one of them, such as direct, indirect, coreference, and ellipsis questions. We stick to one dataset in experiments due to the following reasons 1) all the multi-task learning framework has been trained and tested only on the CSQA dataset. Hence, for a fair evaluation and comparison of our approach inheriting the evaluation settings same as [19,21], we stick to the CSQA dataset. 2) other approaches [2,24] on datasets such as ConvQA, TREC CAsT, etc are not multi-task learning approaches. Further, we cannot retrain [19,8,21] on these datasets due to their missing logical forms employed by each of these models.

We incorporate a semi-automated preprocessing step to annotate the CSQA dataset with gold logical forms. For each question type and subtype in the dataset, we create a general template with a pattern sequence that the actions should follow. Thereafter, for each question, we follow a set of rules to create the specific gold logical form that extracts the gold sequence of actions based on the type of the question. The actions used for this process are the one in Table 2.

**CARTON Configurations**  For the transformer network, we use the configurations from [25]. Our model dimension is $d_{model} = 512$, with a total number of $H = 8$ heads and layers $L = 4$. The inner feed-forward linear layers have dimension $d_{ff} = 2048$, (4 * 512). Following the base transformer parameters, we apply residual dropout to the summation of the embeddings and the positional encodings in both encoder and decoder stacks with a rate of 0.1. On the other hand, the pointer networks also use a dropout layer for the linear projection of the knowledge graph embeddings. For predicates and types, we randomly initialize the embeddings and are jointly learned during training. The KG embeddings dimension of predicate and types match the transformer model

---

[7] https://amritasaha1812.github.io/CSQA

dimension, $d_{kg} = 512$. However, for the entities, we follow a different initialization. Due to a significantly high number of the entities, learning the entity embeddings from scratch was inefficient and resulted in poor performance. Therefore, to address this issue, we initialized the entity embeddings using sentence embeddings that implicitly use underlying hidden states from BERT network [3]. For each entity, we treat the tokens that it contains as a sentence, and we feed that as an input. We receive as output the entity representation with a dimension $d_{ent} = 768$. Next, we feed this into a linear layer that learns, during training, to embed the entity into the same dimension as the predicates and types.

**Models for Comparison**   To compare the CARTON framework, we use the last three baselines that have been evaluated on the employed dataset. The authors of the CSQA dataset introduce the first baseline: HRED+KVmem [19] model. HRED+KVmem employs a seq2seq [23] model extended with memory networks [22,12]. The model uses HRED model [20] to extract dialog representations and extends it with a Key-Value memory network [17] for extracting information from KG. Next, D2A [8] uses a semantic parsing approach based on a seq2seq model, extended with dialog memory manager to handle different linguistic problems in conversations such as ellipsis and coreference. Finally, MaSP [21] is the current state-of-the-art model and is also a semantic parsing approach. It is a multi-task framework with entity detection, based on the transformer architecture.

**Evaluation Metrics**   To evaluate CARTON, we use the same metrics as employed by the authors of the CSQA dataset [19] and previous baselines. We use the "F1-score" for questions that have an answer composed by a set of entities. "Accuracy" is used for the question types whose answer is a number or a boolean value (YES/NO).

## 5   Results

We report our empirical results in Table 4, and conclude that CARTON outperforms baselines average on all question types (row "overall" in the table). We dig deeper into the accuracy per question type to understand the overall performance. Compared to the current state-of-the-art (MaSP), CARTON performs better on eight out of ten question types. CARTON is leading MaSP in question type categories such as *Logical Reasoning (All)*, *Quantitative Reasoning (All)*, *Simple Question (Coreferenced)*, *Simple Question (Direct)*, *Simple Question (Ellipsis)*, *Verification (Boolean)*, *Quantitative Reasoning (Count)*, and *Comparative Reasoning (Count)*. Whereas, MaSP retains the state of the art for the categories of *Clarification* and *Comparative Reasoning (All)*. The main reason for weak results in *Comparative Reasoning (All)* is that our preprocessing step finds limitation in covering this question type and is one of the shortcoming of our proposed grammar[8]. We investigated several reasonable ways to cover *Comparative Reasoning (All)* question type. However, it was challenging to produce a final answer set identical to

---

[8] For instance, when we applied the preprocessing step over the test set, we could not annotate the majority of the examples for the *Comparative Reasoning (All)* question type.

**Table 4.** Comparisons among baseline models on the CSQA dataset having 200K dialogues with 1.6M turns, and over 12.8M entities.

| Methods | HRED-KV | D2A | MaSP | CARTON (ours) | $\Delta$ |
|---|---|---|---|---|---|
| **Question Type (QT)** | | | **F1 Score** | | |
| Overall | 9.39% | 66.70% | 79.26% | **81.35%** | **+2.09%** |
| Clarification | 16.35% | 35.53% | **80.79%** | 47.31% | -33.48% |
| Comparative Reasoning (All) | 2.96% | 48.85% | **68.90%** | 62.00% | -6.90% |
| Logical Reasoning (All) | 8.33% | 67.31% | 69.04% | **80.80%** | +11.76% |
| Quantitative Reasoning (All) | 0.96% | 56.41% | 73.75% | **80.62%** | +6.87% |
| Simple Question (Coreferenced) | 7.26% | 57.69% | 76.47% | **87.09%** | +10.62% |
| Simple Question (Direct) | 13.64% | 78.42% | 85.18% | **85.92%** | +0.74% |
| Simple Question (Ellipsis) | 9.95% | 81.14% | 83.73% | **85.07%** | +1.34% |
| **Question Type (QT)** | | | **Accuracy** | | |
| Overall | 14.95% | 37.33% | 45.56% | **61.28%** | **+15.72%** |
| Verification (Boolean) | 21.04% | 45.05% | 60.63% | **77.82%** | +17.19% |
| Quantitative Reasoning (Count) | 12.13% | 40.94% | 43.39% | **57.04%** | +13.65% |
| Comparative Reasoning (Count) | 8.67% | 17.78% | 22.26% | **38.31%** | +16.05% |

the gold answer set. For instance, consider the question *"Which administrative territories have diplomatic relations with around the same number of administrative territories than Albania?"* that includes logic operators like "around the same number", which is ambiguous because CARTON needs to look for the correct answer in a range of the numbers. Whereas, MaSP uses a BFS method to search the gold logical forms and performance is superior to CARTON. The limitation with *Comparative Reasoning* question type also affects CARTON's performance in the *Clarification* question type where a considerable number of questions correspond to *Comparative Reasoning*. Based on analysis, we outline the following two reasons for CARTON's outperformance over MaSP: First, the MaSP model requires to perform entity recognition and linking to generate the correct entity candidate. Even though MaSP is a multi-task model, errors at entity recognition step will still be propagated to the underlying coreference network. CARTON is agnostic of such a scenario since the candidate entity set considered for each conversation turn is related to the entire relevant context (the previous question, answer, and current question). In CARTON, entity detection is performed only by stacked pointer networks. Hence no error propagation related to entities affects previous steps of the framework. Second, CARTON uses better supervision signals than MaSP. As mentioned earlier, CARTON supervision signals propagate in sequential order, and all components use the signal forwarded from the previous components. In contrast, the MaSP model co-trains entity detection and semantic parsing with different supervision signals.

### 5.1   Ablation Study

An ablation study is conducted to support our architectural choices of CARTON. To do so, we replace stacked pointer networks module with simple classifiers. In particular,

**Table 5.** CARTON ablation study. "W/o St. Pointer" column shows results when stacked pointers in CARTON is replaced by classifiers.

| Question Type (QT) | CARTON | W/o St. Pointers |
|---|---|---|
| Clarification | 47.31% | 42.47% |
| Comparative Reasoning (All) | 62.00% | 55.82% |
| Logical Reasoning (All) | 80.80% | 68.23% |
| Quantitative Reasoning (All) | 80.62% | 71.59% |
| Simple Question (Coreferenced) | 87.09% | 85.28% |
| Simple Question (Direct) | 85.92% | 83.64% |
| Simple Question (Ellipsis) | 85.07% | 82.11% |
| Verification (Boolean) | 77.82% | 70.38% |
| Quantitative Reasoning (Count) | 57.04% | 51.73% |
| Comparative Reasoning (Count) | 38.31% | 30.87% |

**Table 6.** CARTON stacked pointer networks results for each question type. We report CARTON's accuracy in predicting the KG items such as entity, predicate, or type.

| Question Type (QT) | Entity | Predicate | Type |
|---|---|---|---|
| Clarification | 36.71% | 94.76% | 80.79% |
| Comparative Reasoning (All) | 67.63% | 97.92% | 77.57% |
| Logical Reasoning (All) | 64.7% | 83.18% | 91.56% |
| Quantitative Reasoning (All) | - | 98.46% | 73.46% |
| Simple Question (Coreferenced) | 81.13% | 91.09% | 80.13% |
| Simple Question (Direct) | 86.07% | 91% | 82.19% |
| Simple Question (Ellipsis) | 98.87% | 92.49% | 80.31% |
| Verification (Boolean) | 43.01% | 94.72% | - |
| Quantitative Reasoning (Count) | 79.60% | 94.46% | 79.51% |
| Comparative Reasoning (Count) | 70.29% | 98.05% | 78.38% |

predicates and types are predicted using two linear classifiers using the representations from the contextual encoder. Table 5 illustrates that the modified setting (w/o St. Pointers) significantly under-performs compared to CARTON in all question types. The stacked pointer networks generalize better in the test set due to their ability to learn meaningful representations for the KG items and align learned representations with the conversational context. While classifiers thoroughly learn to identify common patterns between examples without incorporating any information from the KG. Furthermore, our framework's improved results are implied from the ability of stacked pointer networks to handle out-of-vocabulary entities, predicates, and types that are unseen during training.

## 5.2   Error Analysis

We now present a detailed analysis of CARTON by reporting some additional metrics. Table 6 reports the accuracy of predicting the KG items such as entity, predicate, or type using CARTON. The prediction accuracy of KG items is closely related to the performance of our model, as shown in Table 4. For example, in the *Quantitative (All)* question type (Table 4), predicting the correct type has an accuracy of $73.46\%$ which is lowest compared to other question types. The type prediction is essential in such category of questions, where a typical logical form possibly is: *"argmin, find_tuple_counts, predicate, type1, type2"*. Filtering by the wrong type gets the incorrect result. Please note, there is no "entity" involved in the logical forms of *Quantitative (All)* question type. Hence, no entity accuracy is reported.

   Another interesting result is the high accuracy of the entities and predicates in *Comparative (Count)* questions. Also, the accuracy of type detection is $78.38\%$. However, these questions' accuracy was relatively low, only $38.31\%$, as reported in Table 4. We believe that improved accuracy is mainly affected due to the mapping process of entities, predicates, and types to the logical forms that is followed to reach the correct answer. Another insightful result is on *Simple (Ellipsis)*, where CARTON has a high entity accuracy compared with *Simple Question*. A possible reason is the short length of the question, making it easier for the model to focus on the right entity. Some example of this question type is *"And what about Bonn?"*, where the predicate is extracted from the previous utterance of the question.

   We compute the accuracy of the decoder which is used to find the correct patterns of the logical forms. We also calculate the accuracy of the logical forms after the pointer networks initialize the KG items. We report an average accuracy across question types for generating logical form (by decoder) as $97.24\%$, and after initializing the KG items, the average accuracy drops to $75.61\%$. Higher accuracy of logical form generation shows the decoder's effectiveness and how the Transformer network can extract the correct patterns given the conversational context. Furthermore, it also justifies that the higher error percentage is generated while initializing the items from the KG. When sampling some of the wrong logical forms, we found out that most of the errors were generated from initializing a similar predicate or incorrect order of the types in the logical actions.

## 6   Conclusions

In this work, we focus on complex question answering over a large-scale KG containing conversational context. We used a transformer-based model to generate logical forms. The decoder was extended with a stack of pointer networks in order to include information from the large-scale KG associated with the dataset. The stacked pointer networks, given the conversational context extracted by the transformer, predict the specific KG item required in a particular position of the action sequence. We empirically demonstrate that our model performs the best in several question types and how entity and type detection accuracy affect the performance. The main drawback of the semantic parsing approaches is the error propagated from the preprocessing step, which is not 100% accurate. However, to train the model in a supervised way, we need the gold logical form annotations. The model focuses on learning the correct logical forms, but there is

no feedback signal from the resulting answer generated from its logical structure. We believe reinforcement learning can solve these drawbacks and is the most viable next step of our work. Furthermore, how to improve the performance of Clarification and Comparative Reasoning question type is an open question and a direction for future research.

## Acknowledgments

The project leading to this publication has received funding from the European Union's Horizon 2020 research and innovation program under the Marie Skłodowska-Curie grant agreement No. 812997 (Cleopatra).

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
