# OpenReview forum: "Context Transformer with Stacked Pointer Networks for Conversational Question Answering over Knowledge Graphs"
_eswc-conferences.org/ESWC/2021/Conference/Research_Track — ESWC 2021 Research_

### Official Review · AnonReviewer4 · 2021-01-12
**The paper shows how additions to a framework improve overall performance on a common task in the field. Overall presentation is good, but could be more clearly in certain areas.**

**Rating:** 1
**Confidence:** 4
**Impact:** 3
**Design And Technical Quality:** 3

**Review:**

The authors propose a novel framework for conversational question answering which implements two improvements over the current SOTA:
1.	In addition to the conversational context, the framework also uses the underlying KG information
2.	Every module of the proposed framework makes use of the signals produced in the previous components, as opposed to approaches that use only the signal of one preceding module

Furthermore, the authors provide a generally usable grammar for generating logical forms.
The proposed framework consists of three modules (and a KGE module? – see weak points), namely the 1) contextual encoder, 2) logical decoder and the 3) stacked pointer network. The framework can be summarized as follows:
The contextual encoder takes the word embeddings of the textual sequence as input and outputs a representation of the current context. The logical decoder takes the current context from 1) and outputs the logical form of the input sequence. Finally, the stacked pointer network takes the contextual representation, the logical form (and the knowledge graph embeddings?) and outputs the candidates for the entity, type and predicate to fill the logical form and query against the underlying knowledge graph.
Here, both the usage of knowledge graph embeddings and the forward passing of the contextual information to the pointer stack are novel compared to the state of the art.

Evaluation of the proposed framework is done using one dataset (Complex sequential Question Answering), because other multi-task learning approaches also used this dataset, and other datasets differ in their underlying task.
The main findings from the experimental sections are:
1.	an improvement over SOTA approaches
2.	that incorporating the KG significantly adds to this improvement

Overall, the paper shows how the authors implemented the usage of additional information to the process of conversational question answering over knowledge graphs and that these changes lead to an improvement. For the most part, the presentation is clear (see weak points for an exception) and understandable. The ablation study in the experimental section shows that the proposed changes significantly contribute to the improvement in evaluation. Both the data and the implementation are available on Github, thus providing for reusability, along with the presented experimental settings.


**Anonymity:**

Yes, I would like my review to remain anonymous.

**Reuse And Availability:**

4: High

**Strong Points:**

+ empirical evaluation shows an overall improvement of the proposed approach in comparison with SOTA approaches
+ an ablation study shows that the improvement is caused by incorporating the underlying KG, thus supporting the author’s claims


**Subreviewer:**

I delegated this review to a subreviewer.

**Weak Points:**

In some parts, the presentation is unclear:
- were the word embeddings trained on the conversational data, or was external data used?
- since the knowledge graph embeddings are jointly trained with the rest of the framework, it is confusing that this is not shown in Fig. 1 or mentioned in Section 3.2, but only mentioned in the training section. Is the KGE part not considered as a part of the overall architecture?

---

> ### Author Rebuttal · Authors · 2021-01-27
>
> We thank the reviewer for carefully reviewing our submission and for the thoughtful comments and feedback. We would like to address these and provide further details on two main points:
>
> 1) Were the word embeddings trained on the conversational data, or was external data used?
>
>     Answer: We initialise word embeddings of our input utterance using GloVe. We do not pre-train them on conversational data. Those embeddings are used as input to our transformer encoder.
>
>
> 2) Is the KGE part not considered as a part of the overall architecture?
>
>     Answer: The knowledge graph embeddings are part of the overall architecture; we will also clarify this in detail in section 3.2. For simplicity, we did not add it on the figure; we will consider updating the figure upon acceptance.

---

> > ### Comment · AnonReviewer4 · 2021-01-31
> > **Thanks for the rebuttal**
> >
> > I read and acknowledge the rebuttal. No further questions.

---

### Official Review · AnonReviewer2 · 2021-01-14
**The authors of this paper propose the framework CARTON, which performs multi-task semantic parsing for the task of complex question answering over large-knowledge graphs.**

**Rating:** 2
**Confidence:** 5
**Impact:** 4
**Design And Technical Quality:** 4

**Review:**

This is a well-written paper about a new multi-task framework for conversational question answering over large knowledge graphs. Firstly, the authors predefined a grammar with actions that can result in different logical forms, which can be executed on the KG. In section 5, the authors provide empirical evidence to prove that these actions can create suitable sequences that cover complex questions. The semantic parsing part of CARTON was performed with a context transformer, and the decoder was extended with a stack of pointer networks. Based on the ablation study conducted by the authors, this stacked pointer network is proven to be a very useful component of the framework. Moreover, the authors provide a detailed evaluation of CARTON on CSQA, in comparison with the three baselines evaluated in the same dataset. Last, apart from the ablation study, the authors provide a detailed error analysis of CARTON as well, which makes their paper more complete.

Some minor typos: “empirical evidences” to “empirical evidence”, “are the one in Table 2” to “are the ones in Table 2”.

I have read the author's rebuttal which does not change my positive view of the paper.




**Anonymity:**

Yes, I would like my review to remain anonymous.

**Reuse And Availability:**

4: High

**Strong Points:**

* Compared to the current state-of-the-art (MaSP), the authors proved that CARTON performs better on eight out of ten question types. Regarding CARTON's outperformance over MaSP, they commented that in CARTON, entity detection is performed only by the stacked networks, so there is no error propagation, which may occur in the MaSP model. Moreover, the authors explained that CARTON uses better supervision signals than MaSP, which propagate in sequential order, while the MaSP model uses different supervision signals to co-train entity detection and semantic parsing.

* The authors conducted an ablation study to support their architectural choices of CARTON. They concluded that the stacked pointer networks generalize well, so they proved their ability to not only learn meaningful representations for KG items, but also align these learned representations with the conversational context.

* The authors analysed some drawbacks their framework has, such as the weak results in Comparative Reasoning, and they proposed ways to overcome them in the future.


**Subreviewer:**

I delegated this review to a subreviewer.

**Weak Points:**

* In Section 4, the authors mention that they initialized the entity embeddings using sentence embeddings that implicitly use underlying hidden states from BERT. It would be useful to explain this choice more, as there are many different representations an entity can have based on these hidden states. For example, did you use the embeddings of the final layer of BERT, or did you use something more sophisticated?

---

> ### Author Rebuttal · Authors · 2021-01-27
>
> We thank the reviewer for carefully reviewing our submission and for the thoughtful comments and feedback. We would like to address these and provide further details on one main point:
>
> 1) Regarding BERT embeddings for entities.
>
>     Answer: We use the final layer of the BERT network as initial representations for the entities. Those embeddings are fed into a linear layer that learns, during training, to project the entity into the same dimension as the predicates and types. We will further clarify this with more details upon acceptance.

---

### Official Review · AnonReviewer1 · 2021-01-14
**Review for QA paper over CSQA**

**Confidence:** 4
**Impact:** 2
**Design And Technical Quality:** 4

**Review:**

This paper presents a neuronal network architecture for solving a popular benchmark for conversational question answering over Knowledge graphs. The network has the following architecture:
1) a transformer based encoder
2) a decoder for a set of grammar rules together with tokens for entity, predicate and type placeholders
3) 3 pointer networks with the aim of replacing the placeholders
The input of the network is the last part of the current conversation with the new question, the output is a logical form. The Neuronal networks minimises the correct order of the logical form and the correct mapping of the placeholders.

I have two main problems with this paper.
1) The proposed architecture is very similar to the state of the art system cited by the authors, i.e. MaSP. See https://arxiv.org/pdf/1910.05069.pdf Figure 1. and Table 1 (for the grammar). The difference is rather minimal, and rather on an engineering level in my opinion.
2) The experimental results are comparing, the authors work, with the system MaSP. Besides errors on my side, the authors compare their experimental results, with the lowest results presented in https://arxiv.org/pdf/1910.05069.pdf (see Table 3). The Table 4 in this paper is compared with Table 3 (of MaSP) which contains 3 columns (Vanilla,  BERT ,  Large Beam). But strangely Vanilla is used to compare with. The authors use a BERT encoder so there is no reason to compare with Vanilla. But comparing with Large Beam is in fact the most meaningful. In this case MaSP is still the state-of-the art system. So the argumentation to slightly change the architecture of the network seam not reasonable.
To summarise, the work seams to be incremental over exiting one and the added changes do not seams to have an impact. I therefore would refuse this paper.

Further notes:
- using your grammar rules, how are you able to handle clarifications and verifications? I do not see how you can return something like "Did you mean Robbiate ?". This is apperently possible since you report for clarification an accuracy of 47%. If this is possible you should explain, maybe giving an example in Table 3.
- 3.2 you repeat x as symbols, once as tokens for the question and one for the embedding
- how do you extract types? this is unclear to me since you are doing it over wikidata
- how do you select the candidates for the variable vocabulary in the pointer network responsible for the entity.
- 4 point 1 is not really a good argument. I think the real argument is what you say after, i.e. we cannot retrain on these datasets due to their missing logical forms. This is argument enough.
- it is not so clear how you generate the logical forms, I thought they are provided by the benchmmark. Or do you mean adapt their logical forms to the grammar you choose?
- Therefore, address this issue, we in ..... attention grammar
- 5 results are public in Github -> no link, I could not find it
- Table 5, I would add the row overall
- there is no documentation in the github repo

**Anonymity:**

Yes, I would like my review to remain anonymous.

**Rating:**

-2: Reject

**Reuse And Availability:**

3: Medium

**Strong Points:**

- nice benchmark

**Subreviewer:**

I submitted this review.

**Weak Points:**

- there is a very similar work
- the code is open source but no documentation is available which makes it problematic to reproduce the code
- there seams to be no improvement over the state of the art

---

> ### Author Rebuttal · Authors · 2021-01-27
>
> We thank the reviewer for carefully reviewing our submission and for the thoughtful comments and feedback. We would like to address these and provide further details on nine main points:
>
> 1) The proposed architecture is very similar to the state of the art system (MaSP).
>
>     Answer: The main methodological differences between our framework and Shen et al. (2019) are the following (which we do not just consider an engineering effort):
>     * CARTON’s stacked pointer networks incorporate knowledge graph information for performing reasoning and do not rely only on the conversational context as MaSP does.
>     * The stacked pointer network architecture is used intentionally to provide the flexibility for handling out-of-vocabulary entities, predicates, and types that are unseen during training. Our ablation study further supports our choice. This is never covered by MaSP as their model never intended to have this flexibility in it.
>     * Another methodological novelty is that the supervision signals in CARTON are propagated in sequential order, and all the components use the signal forwarded from the previous component. On the other hand, MaSP has independent modules which receive “different” supervision signals, and they only co-train them under a multi-task strategy.
>     * MaSP authors employ almost the same grammar with the D2A model. However, in our work, we provide a new semantic grammar for generating logical forms with new actions that D2A grammar does not have.
>
>
> 2) The experimental results are compared, the authors work, with the system MaSP.
>
>     Answer: We use the final layer of the BERT network only for initialising the entities before our entity pointer makes the prediction for the logical form. We do not use a BERT encoder for the framework; we use a vanilla transformer encoder which is trained from scratch. We also do not employ any beam search on our experiments; therefore, the model we present in the paper is a vanilla version. We stick to vanilla transformer because replacing them with BERT or RoBERTA is an engineering effort and we do not consider it as a scientific contribution (Same applies for beam search).
> Our vanilla model still has state of the art results on seven out of ten (for MaSP-BERT) and six out of ten (for MaSP-Beam search) question types respectively (please note that the overall score is weighted hence simple questions heavily affect it, comparing question types individually it is a more fair way of judging the model). We agree that this has not been clearly explained in the paper.
>
>
>
> 3) Regarding the grammar rules for clarification.
>
>     Answer: None of the existing approaches defines a specific rule for clarification question type. For clarification, we consider the complete question as input. Where we also include the clarification part on it. This is similar to how previous works have treated this question type. For instance, considering Table 1 in our paper, for the clarification part, our framework would consider as input both turn 2 (T2) and turn 3 (T3).
> For verification question type, we provide an example in Table 3, and we will also include one for clarification upon acceptance.
>
>
> 4) On section 3.2 you repeat x as symbols, once as tokens for the question and one for the embedding.
>
>     Answer: Thank you for pointing that out, we will take care to fix it for the final version.
>
>
> 5) How do you extract types? This is unclear to me since you are doing it over Wikidata.
>
>     Answer: We do not extract types; they are given for each question from the CSQA dataset since it was also built on top of Wikidata. Our (entity) type pointer network is responsible for predicting the corresponding type for each question. In subsection “Stacked Pointer Networks” we provide more details on how it works.
>
>
> 6) How do you select the candidates for the variable vocabulary in the pointer network responsible for the entity.
>
>     Answer: We clarify this in subsection Stacked Pointer Networks.  “In each conversation, we predict a probability distribution only for the entities that are part of the context...”
>
>
> 7) It is not so clear how you generate the logical forms, I thought they are provided by the benchmark. Or do you mean adapt their logical forms to the grammar you choose?
>
>     Answer: No, the logical forms are not provided by the benchmark. Since we introduce a new grammar, we had to incorporate a semi-automated preprocessing step to annotate the CSQA dataset with gold logical forms. We also provide the code to annotate the dataset on our GitHub repository.
>
>
> 8) Table 5, I would add the row overall.
>
>     Answer: Yes, we can add it upon acceptance.
>
>
> 9) There is no documentation in the GitHub repo.
>
>     Answer: ~~We are preparing detailed documentation of our GitHub repository.~~ A first version of the documentation is currently available. The documentation covers steps on annotating CSQA dataset using our grammar, and how to execute our framework.

---

### Decision · Program_Chairs · 2021-02-23

**Decision:**

Accept with shepherding

**Comment:**

This meta review summarises the strengths and weaknesses pointed out by the reviewers. There was consensus among the reviewers that the paper addresses an important problem within scope of the conference. The reviewers also agreed that the evaluation has adopted a rigorous protocol by using a large set of benchmarks and ablation analysis. However, as evident from the initial reviews, there remain some issues that need addressing. These primarily relate to the lack of a clear comparison against one state-of-the-art method.  For the paper to be accepted, the authors are kindly asked to address the following issues and submit a revised version of the paper:

[Task 1] Revise the related work section by making an explicit comparison against MaSP, clearly identifying the similarities and differences. Authors should clarify which components of MaSP are reused, which are replaced, and which are extended. Currently, the last part of the first paragraph on page 4 (starting with 'Finally, [22]...') is not a sufficient comparison to MaSP. Please clearly acknowledge the similarities of your method to MaSP.

[Task 2] The components of the method proposed in this work and the structure of the content in which it is presented is similar to the MaSP paper [22]. Authors should make explicit contrast to MaSP when similar components are introduced in their methodology section. For example, where the architecture is introduced and it looks similar, explain the main differences.

[Task 3] In the introduction, the authors claim that the novelty being '... incorporate knowledge graph information for performing any reasoning and does not rely only on the conversational context. ... provide the flexibility for handling out-of-vocabulary [27] entities, predicates, and types that are unseen during training... the supervision signals in CARTON propagate in sequential order, and all the components use the signal forwarded from the previous components'. Please explicitly clarify how these are addressed by your method. It may be appropriate to add explanations at different parts of your methodology, where corresponding components are introduced.

Please submit the revised version of the paper by March 17th 2021. Should you have any questions or want feedback on your changes, feel free to contact Ziqi Zhang (ziqi.zhang@sheffield.ac.uk).